# Extracellular Vesicles Cargo in Modulating Microglia Functional Responses

**DOI:** 10.3390/biology11101426

**Published:** 2022-09-29

**Authors:** Maria Ester La Torre, Maria Antonietta Panaro, Melania Ruggiero, Rita Polito, Antonia Cianciulli, Francesca Martina Filannino, Dario Domenico Lofrumento, Laura Antonucci, Tarek Benameur, Vincenzo Monda, Marcellino Monda, Chiara Porro, Giovanni Messina

**Affiliations:** 1Department of Clinical and Experimental Medicine, University of Foggia, 71122 Foggia, Italy; 2Department of Biosciences, Biotechnologies and Biopharmaceutics, University of Bari, 70125 Bari, Italy; 3Department of Biological and Environmental Sciences and Technologies, Section of Human Anatomy, University of Salento, 73100 Lecce, Italy; 4Department of Biomedical Sciences, College of Medicine, King Faisal University, Al-Ahsa 31982, Saudi Arabia; 5Department of Experimental Medicine, Section of Human Physiology and Unit of Dietetics and Sports Medicine, University of Campania “Luigi Vanvitelli”, 80138 Naples, Italy

**Keywords:** extracellular vesicles, LPS, microglia, BV2, migration, microglia polarization, inflammation

## Abstract

**Simple Summary:**

Extracellular vesicles (EVs) are considered a new additional mechanism of intercellular communication also in brain cells. Microglia are brain-resident immune cells that provided to immune surveillance and inflammatory responses in the central nervous system. The goal of this research is to investigate how EVs of brain, in particular if EVs isolated from microglia in response to LPS (Lipopolysaccharide) may have the ability to induce an inflammatory state in microglia. This study has showed that EVs with LPS cargo can activate microglia in a manner similar to that of LPS alone and that EVs derived from control cells cannot polarize microglia towards a pro-inflammatory state. The importance of this study is the demonstration that EVs produced in an inflammatory environment can exacerbate the inflammatory message by activating microglia, which may have a negative impact on all brain cells.

**Abstract:**

Extracellular vesicles (EVs) represent a heterogeneous group of membranous structures derived from cells that are released by all cell types, including brain cells. EVs are now thought to be an additional mechanism of intercellular communication. Both under normal circumstances and following the addition of proinflammatory stimuli, microglia release EVs, but the contents of these two types of EVs are different. Microglia are considered the brain-resident immune cells that are involved in immune surveillance and inflammatory responses in the central nervous system. In this research, we have analyzed the effects of EVs isolated from microglia in response to LPS (Lipopolysaccharide) on microglia activation. The EVs produced as result of LPS stimulation, knows as EVs-LPS, were then used as stimuli on microglia BV2 resting cells in order to investigate their ability to induce microglia to polarize towards an inflammatory state. After EVs-LPS stimulation, we analyzed the change to BV2 cells’ morphology, proliferation, and migration, and investigated the expression and the release of pro-inflammatory cytokines. The encouraging findings of this study showed that EVs-LPS can activate microglia in a manner similar to that of LPS alone and that EVs derived from control cells cannot polarize microglia towards a pro-inflammatory state. This study has confirmed the critical role of EVs in communication and shown how EVs produced in an inflammatory environment can exacerbate the inflammatory process by activating microglia, which may have an impact on all brain cells.

## 1. Introduction

Microglia are resident macrophages of the central nervous system (CNS) that account for 10% of all cells in the brain [1,2]. Microglia are associated with the pathogenesis of many neurodegenerative diseases and other inflammatory diseases in the brain. They help nourish and support neurons, clear debris, and respond to foreign stimuli [3]. Microglia exists in two states: resting and activated [4]. Microglia are at rest in the healthy brain, where they use highly motile ramified processes to sample the neural parenchyma every few hours. Microglia undergo morphological and functional changes when brain homeostasis is disrupted by infection, trauma, or neurodegeneration, a process known as microglial activation [5]. In terms of morphology, activation causes a graded response of decreased arborization, enlargement of the cell soma, and shortening or loss of cellular processes. Reactive microglia travel to the sites of lesion or infection and undergo mitotic proliferation, increasing in density in order to provide additional defense and tissue homeostasis restoration [6]. Microglia may exist into two opposite types—classical (M1) or alternative (M2)—and can transition from one phenotype to another. Microglial cells are activated by infection or injury towards the pro-inflammatory phenotype (M1), which produces pro-inflammatory mediators and induces inflammation and neurotoxicity. In addition, an alternative anti-inflammatory phenotype (M2) may emerge in which microglial cells release anti-inflammatory mediators and induce anti-inflammatory behavior and neuroprotectivity. Different studies have demonstrated that balancing microglia M1/M2 polarization had a promising therapeutic prospect in neurodegenerative diseases [7]. Microglia lose or change their homeostatic function in neurodegenerative diseases, resulting in neuroinflammation. As microglia-mediated neuroinflammation is a common feature shared by various neurodegenerative diseases, including Alzheimer’s disease (AD), Parkinson’s disease (PD), amyotrophic lateral sclerosis (ALS), and multiple sclerosis (MS) [8], recent research studies have focused on the molecular pathways that lead to microglia activation. Extracellular vesicles (EVs) are considered a new emerging mode of intercellular communication due to their ability to carry and deliver various cargos such as cytosolic proteins, growth factors, cytokines/chemokines, lipids, RNA, and miRNA that can be transferred to recipient cells [9,10]. EVs are classified based on their cellular origin and biological function, as well as their biogenesis. Cells produce different classes of EVs with a diameter of 1–1000 nm: exosomes (30–100 nm), microvesicles (MVs) (100–2000 nm), and apoptotic bodies (1–5 nm). Also, nanovesicles that are similar in size to exosomes and released by budding off the plasma membrane like MVs, are defined as exosomes, therefore it will be better to use the more general term “EVs” [11,12]. EVs are released by a variety of cells, including immune cells, and can serve as antigen-presenting vesicles, stimulants of anti-humoral immune responses, or inducers of tolerogenic effects to suppress inflammation [9,13]. EVs are involved in various pathophysiological processes. EVs contribute to intercellular communication in the brain through their basal release and being taken up by neighboring cells, or by releasing into the cerebrospinal fluid (CSF) and blood [14]. The role of EVs released by activated microglia in neuroinflammation remains poorly understood. Lipopolysaccharide (LPS) is a common inflammatory trigger widely used to induce and study neuroinflammation. It is known that LPS expressed in the external membrane of Gram-negative bacteria can activate microglial cells and thereby promote neuroinflammatory processes by secreting different types of cytokines and eicosanoids [15]. As a result, LPS-induced BV2 microglial cells are frequently used as a model to evaluate the therapeutic potential of different candidates to prevent neuroinflammation [16,17]. In this study, we hypothesized that stimulation of BV2 microglial cells with LPS induces the release of EVs containing pro-inflammatory molecules that may contribute to the spread of inflammation in the brain parenchyma. Our findings shed light on the role of EVs in microglia cell communication and the spread of neuroinflammation in the brain when taken together. The EVs obtained with LPS stimulation were used as stimuli on resting BV2 microglial cells in order to investigate their ability to induce microglial polarization towards an inflammatory state. In this study, we compared the morphological changes of BV2 cells in response to stimulation with EVs released by different subsets of microglia (resting and activated), and we evaluated proliferation and migration as well as the expression and release of pro-inflammatory cytokines under the same experimental conditions. Taken together, our findings shed light on the role of EVs in microglia cell communication and the spread of neuroinflammation in the brain.

## 2. Materials and Methods

### 2.1. Cell Culture and Treatments

The murine BV2 microglia cell line, generated from C57/BL6 female mouse was purchased from the American Type Culture Collection (Manassas, VA, USA). Cells (2 × 10⁵ cells/mL) were grown in Dulbecco’s modified Eagle’s medium (DMEM; High Glucose; Euroclone; Milano) supplemented with 10% fetal bovine serum (FBS; Euroclone; Milan), 100 units/mL penicillin, 100 μg/mL streptomycin, (Penicillin—Streptomycin; Euroclone; Milan, Italy), and 2 mM glutamine (Glutamine; Euroclone; Milan) at 37 °C in a humidified atmosphere of 95% air and 5% carbon dioxide (CO2) until 80% confluence and then subsequently trypsinized with Trypsin-EDTA (Trypsin-EDTA 1X in PBS). For the cell migration and morphological tests, cells were used in the fourth cell passage and were seeded in six-well plates, while they were seeded into 24-well plates for the cell viability assay. The cells were then treated with LPS from Escherichia coli (O128: B12; Sigma-Aldrich) (LPS) (1 µg/mL). Extracellular vesicles from resting BV2 cells (EVs-CTR), and extracellular vesicles obtained from cells stimulated with LPS (EVs-LPS) at the protein concentration of 10 µg/mL [18,19].

### 2.2. EVs Production

The murine BV2 microglial cell line was used for EVs production. Cells were seeded and cultured as previously described [20] until they reached 80% confluence, and then they were stimulated with LPS (1 ug/mL) for 24 h. After incubation, supernatants of unstimulated and stimulated cells were obtained by centrifugation at 1500× *g* for 10 min to remove cells and large debris, respectively. After successive centrifugation steps (45 min at 14,000× *g*), EVs from the supernatants were washed and recovered in 400 µL NaCl (0.9 percent *w*/*v*). As a control, the last supernatant’s washing medium was used (vehicle). Determination of the amount of EVs was carried out by measuring total EV-associated proteins; proteins were quantified with the Qubit Protein Assay Kit by Qubit^®^2.0Fluorometer (Life Technologies, Eugene, OR, USA).

### 2.3. Assessment of Morphological Changes of BV2 Cells

BV2 microglial cells (5 × 10⁵ cells per well) were incubated overnight in 6-well plates at 37 °C in appropriate culture medium and then treated for 24 h with EVs-CTR, EVs-LPS at the concentration of 10 μg/mL, and LPS at the concentration of 1 μg/mL LPS from Escherichia coli O128: B12; Sigma-Aldrich). The morphological changes of BV2 cells in each group were observed using a light microscope from Leica Microscopy (DM IRB Leica Microsystems GmbH, Wetzlar, German), using 10× and 20× magnification and acquired by the software Leica Application Suite X.

### 2.4. Wound Healing Assay

Approximately 5 × 10⁵ BV2 cells per well were seeded in 6-well plates and grown for 5 days to confluence (100%) at 37 °C and 5% CO_2_ in an appropriate culture medium; the medium was replaced every two days. After 5 days, the monolayer was wounded (wound of 0.5 cm) with a sterile pipet tip to make a 0.5 cm incision. Cells detached upon wounding were carefully rinsed off. Once the wound was made, the medium with the respective stimuli was added, i.e., EVs-CTR (10 µg/mL), and EVs-LPS (10 µg/mL), and LPS (1 ug/mL). The wells were photographed at time 0 (in the moment of cutting) and time 1 (after 24 h). The images were captured with Leica Microscopy (DM IRB Leica Microsystems GmbH, Wetzlar, German), using 10× and 20× magnification and processed with Leica Application Suite software X. All migration tests were performed in triplicate. Migration into the opened wound was photographed with microphotographs after 24 h. The number of migrating cells was quantified by counting all the cells present in the width of the wounds over 24 h using Image J software [21].

### 2.5. Cell Viability Assay

The MTT test was used to measure cellular metabolic activity as an indicator of cell viability, proliferation, and cytotoxicity (Sigma-Aldrich; St. Louis; CAS: 298-93-1) in 24-well plates, approximately 20 × 10⁴ cells were seeded in each well for 24 h. The cells were then treated with different stimuli for 24 h, including EVs-CTR (10 µg/mL), EVs-LPS (10 µg/mL), and LPS (1 µg/mL) for 24 h. Following various treatments, the medium was replaced with 0.5 mg/mL of MTT and incubated for 4 h at 37 °C in 5% CO₂. The formazan crystals were extracted at the end of the incubation period using a cell solubilizing solution (DMSO); 1 mL of DMSO was added to each well and thoroughly stirred for 30 min. A Filter Max F5 Multi-Mode Microplate Reader was used to measure absorbance at 595 nm. Cell viability was measured in percentages (%) as compared to untreated cells, which were assumed to be 100% viable.

### 2.6. Cytokines Assay

The ELISA test was performed using the supernatants from the morphological test. After 24 h, the culture medium was removed and centrifuged at 4 °C for 10 min at 1500 rpm to remove any cellular residues that had accumulated in the medium. For 24 h, cells were incubated at 37 °C and 5% CO_2_ with EVs-CTR, EVs-LPS in a concentration of 10 μg/mL, LPS (1 μg/mL), and control conditions. We evaluated pro-inflammatory cytokines such as TNF-α and anti-inflammatory cytokines such as IL-10 by ELISA test according to the manufacturer’s instructions (R&D system a biotech brand; USA). A microplate reader was used to measure the optical density (OD) at 450 nm (BioTek power wave X5).

### 2.7. Electrophoresis

Following the addition of the stimuli described above, the cells were detached, washed by centrifugation at 600× *g* for 10 min, and lysed in ice-cold lysis buffer [1% (*v*/*v*) Triton X-100, 20 mM Tris-HCl, 137 mM NaCl, 10% (*v*/*v*) glycerol, 2 mM EDTA, 1 mM phenylmethylsulfonyl fluoride (PMSF), 20 µM leupeptin hemisulfate salt, and 0.2 U/mL aprotinin (all from Sigma Aldrich)] for 30 min at 4 °C. The lysates were then centrifuged at 13,800× *g* for 20 min at 4 °C, and the supernatants were adjusted to the same protein concentration (20 µg) by Bradford’s protein assay and separated on SDS-PAGE (NuPage Electrophoresis System-Invitrogen) in NuPage LDS Sample Buffer 4 × 1:4 (*v*/*v*) and with NuPage Sample Reducing Agent 1:10 (500 mM dithiothreitol [DTT] at 10× concentration) on 4–12% Novex Bis-Tris Midi-gel 1.0 mm precast gels (Life Technologies).

### 2.8. Western Blot Analysis

The proteins resolved by electrophoresis were transferred from the gel to nitrocellulose membranes using iBlot Dry Blotting System A (Life-Technologies). Membranes were then blocked with PBS (pH 7.2, containing 0.1% (*v*/*v*) Tween 20 and 5% (*w*/*v*) non-fat dried milk for 1 h, and washed three times with 0.1% Tween 20-PBS (T-PBS) then washed 3 times with 0.1% Tween 20-PBS (T-PBS). Primary antibodies directed against β-actin (1:500), CD40 (1:500), p-Akt (1:500), CD206 (1:500), TLR4 (1:500), and p-IKBα (1:500) (all from Santa Cruz Biotechnology, Inc., Heidelberg, Germany), were incubated for 1 h at room temperature on a shaker and then overnight at 4 °C. At the end, membranes were incubated with horseradish peroxidase (HRP)-conjugated secondary antibodies (Santa Cruz Biotechnology), diluted 1:10,000, for 60 min at room temperature in the dark on a shaker after three washes with 0.1% Tween 20- PBS (T-PBS). Chemiluminescence was used to visualize the immunoreactive bands (BioRad Laboratories, Hercules, CA, USA). Densitometric analysis was performed on the bands obtained after immunoblotting using ID Image Analysis Software (Kodak Digital Science). The β-actin was used as a housekeeping protein for normalizing protein expression levels. Results are expressed as arbitrary units.

### 2.9. Statistical Analysis

All data are presented as a mean ± SD. Statistical analyses were conducted with one-way ANOVA using Graph Prism 9 software (GraphPAD Software, San Diego, CA, USA). *p* < 0.05 was considered to be statistically significant.

## 3. Results

### 3.1. Isolation of Extracellular Vesicles (EVs) Released from Microglial Cells in Different State

Although there is growing evidence that EV released from activated immune cells may play an important role in neuroinflammation, the effects of EVs isolated from microglia cells subjected to inflammatory stimuli on CNS parenchyma have received much less attention. In this study, we isolated EVs from the microglial cell line BV2 in both control cells and in LPS-stimulated cells with 1 μg/mL. Primarily, the cells were grown in the presence and absence of LPS stimuli. After 24 h, EVs were isolated from supernatant medium of the two conditions by a series of centrifugations as described in Material and Methods. The obtained EVs, respectively called EVs-CTR (obtained from unstimulated cell) and EVs-LPS (obtained from LPS stimulated cells), were first counted as previously described [18,19], and then used to stimulate BV2 cells for 24 h in order to assess their biological effects. Figure 1 illustrates the steps of EVs production without (EVs-CTR) and with LPS (EVs-LPS) stimulation of BV2 cells as well as the morphology of BV2 microglia cells stimulated by either EVs-CTR or BV2 cells +LPS (Figure 1).

### 3.2. EVs-LPS Induce Morphological Changes in BV2 Cells

We investigated the morphological changes occurring in microglial activation after treatment with EVs-CTR, EVs-LPS, and LPS in BV2 cells. Normally, LPS caused ramifications loss, expansion of the microglial cell body, and progression toward an amoeboid shape [22]. Figure 2A,C represents the morphology of BV2 cells in the control group and in cells treated with EVs-CTR: it can be seen that cell shape was ramified with a slender and small cell body, whereas after 24 h of stimulation with 1 μg/mL of LPS or EVs-LPS, BV2 cells exhibited an amoeboid shape as indicated by arrows in Figure 2B,D. As shown in the figure, no significant changes were observed in the shape of the cells subjected to EVs-LPS treatment compared to BV2 stimulate with LPS alone.

### 3.3. EVS-LPS Treatment Enhances Cell Migration in BV2 Cell

In response to inflammatory stimuli, neuroinflammation is propagated by the production of pro-inflammatory mediators and the migration of microglial cells toward the infected sites [23]. As a result, we investigated whether extracellular vesicles transporting LPS can induce BV2 cells’ migration. Using wound healing assays as illustrated in Figure 3, we observed that both LPS and EVs-LPS treatment enhances BV2 microglial cell migration when compared to BV2 control cells (*p* < 0.05) or cells treated with EVs-CRT.

### 3.4. Effect of EVs-LPS on BV2 Cell Growth

In order to determine whether EVs influence the viability of BV2 cells, the MTT assay was performed 24 h after treatment with LPS, EVs-LPS, or EVs-CTR. The MTT test findings shown in Figure 4 demonstrated that the cell viability of BV2 cells treated with LPS is significantly (*p* < 0.05) increased in comparison to control cells. Conversely, EVs-LPS-treated BV2 cells exhibited a significant reduction of cell viability in comparison to LPS-treated cells. No significance was observed between control and BV2 treated with EVs.

### 3.5. Effect of EVs-LPS on Cytokines Release

We have evaluated the free concentration of released cytokines from BV2 microglia cells after stimulation with LPS, EVs-LPS, or EVs-CTR. EV-depleted culture medium from BV2 microglia cells was collected and analyzed using ELISA. TNF-α is a pro-inflammatory cytokine produced by microglia that is linked to a neuro-inflammatory picture. TNF-α levels were found to be significantly higher in BV2 cells treated with LPS and EVs-LPS in comparison to control, whereas no significant differences were observed between LPS- and EVs-LPS-treated cells (Figure 5).

We also evaluated the production of IL-10, an anti-inflammatory cytokine that determines the maintenance of the balance of the immune response mediating in neuroprotection. IL-10 levels were significantly higher in BV2 cells in the control and EVs-CTR condition than in the LPS and EVs-LPS conditions, as shown in Figure 6.

### 3.6. Effects of EVs-LPS on the Expression of Other Pro-Inflammatory and Anti-Inflammatory Markers

We evaluated the expression levels of the M1-phenotype CD40 and the M2-phenotype CD206 markers by Western Blot analysis. Results, as reported in Figure 7, show that LPS treatment of BV2 cells induces a significant increase (*p* < 0.05) of CD40 (upper) in comparison to control. Interestingly, we detected a significant increase of CD40 expression in EVs-LPS-stimulated cells compared to LPS-stimulated BV2 cells. Regarding the CD206 evaluation (Figure 7, lower), we observed that both LPS and EVs-LPS treatments were able to increase CD206 expression in comparison to controls, although no significant differences were detected between LPS and EVs-LPS treatments of BV2 cells. Collectively, these results seem to suggest that EVs-LPS are able to drive microglial cells towards a pro-inflammatory response.

The TLR4/Myd88 signaling pathway is directly involved in the inflammatory response induced by LPS, thus we also evaluated whether the signal mediators TLR4 and p-AKT are modulated in terms of protein expression levels by EVs-LPS treatment. The expression of TLR4 did not show significant changes in any of the treatments carried out, as the levels of the protein were comparable in the different experimental tests, as shown in Figure 8 (Panels A and B upper). Regarding the expression of signaling molecules, p-AKT and p-IKBα, we interestingly noted a significant up-regulation of these molecules both in LPS-treated cells and in EVs-LPS-treated cells in comparison to unstimulated cells (Figure 8 Panels A, and B, middle and lower, respectively). Moreover, EVs-LPS treatment was able significantly increase the expression of p-AKT and p-IKBα compared to LPS treatment, thus supporting the idea that EVS-LPS are able to enhance functional pro-inflammatory properties of microglia.

## 4. Discussion

In this study, we investigated the EVs’ ability to activate microglia and to influence the functional state of microglia. For this purpose, LPS was used as a pro-inflammatory stimulus of BV2 microglial cells and caused the release of EVs. Interestingly, the EVs derived from this stimulation, namely EVs-LPS, when used as stimuli on microglia BV2 resting cells determined a change of morphology of BV2 cells, proliferation and migration, although similar to the results obtained after stimulation with LPS alone, whereas EVs derived from control cells cannot activate microglia. Increasing evidence supports the importance of neuroinflammation in neurological diseases such as Alzheimer’s Disease, Parkinson’s Diseases, ischemic stroke, multiple sclerosis, and other diseases. Acute neuroinflammation protects the body by eliminating substances through microglial phagocytosis, whereas chronic neuroinflammation harms the CNS by releasing pro-inflammatory and cytotoxic factors. Microglial cells are CNS resident immune cells that play an important role in the regulation of neural homeostasis as well as the response to injury and repair. Microglia become activated during a pathological state, which leads to microglia proliferation, migration to the site of injury, increased expression of immunomodulators, and transformation into phagocytes capable of clearing damaged cells and debris [24]. Excessive inflammation caused by activated microglia may result in a vicious cycle of neuroinflammation that contributes to neurodegeneration [25]. When activated, microglia undergo dramatic morphologic changes, including cytoskeleton rearrangement, which regulates cell mobility and migration towards the site of inflammation [26,27,28]. The transition of microglia from a resting and ramified phenotype to an amoeboid phenotype is characterized by an enlarged body devoid of ramifications, allowing them to migrate over relatively long distances to accumulate at damage sites [29,30]. These changes are linked to an increase in several transcription factors as well as the release of soluble factors such as proinflammatory molecules [31,32]. All of these processes may result in neuronal damage, which plays an important role in neurodegeneration [33]. Given the consequences of microglia activation, understanding the mechanisms that leads to this, as well as the pathways involved in inhibiting this activation, may be a critical step in the prevention of neurodegenerative diseases. In this study, we evaluated how the microglia–microglia crosstalk could be mediated by EVs. EVs are thought to be in charge of the horizontal transfer of molecules to other cells. Neurons and glial cells in the brain release EVs, leading researchers to believe that EV-mediated communication is a common mechanism in the CNS. EVs play a dual role in in CNS pathologies: on one hand, they maintain cellular homeostasis, cleaning the brain parenchyma of protein aggregates and other pathogenic agents; on the other hand, they contribute to the spread of toxic molecules in the brain, thereby spreading pathogenic messages. The relationship between the cell’s phenotypic state and the content of the EVs released is unknown. Kumar and colleagues have demonstrated that microparticles (MPs) derived from microglia are released and enter the circulation during traumatic brain injury (TBI); additionally, these MPs have in vitro-activated recipient microglia and up-regulated pro-inflammatory molecules [34]. According to Yang (2018), EVs derived from activated microglia contain pro-inflammatory molecules such us TNF-α and IL-6, and on the contrary, there is a low production of anti-inflammatory mediator such as IL-10 [35]. The LPS administration alters the EV cargo, as demonstrated by Jones’s recent study [36]. In this study, the authors found that the number of LPS-stimulated BV-2 produced a higher number of EVs compared to the control, which is consistent with a previous study published by Erickson et al. [37]. In contrast, Chen et al. observed different effects after LPS treatment, since the release of EVs in LPS treated, cardiomyocytes is reduced compared to their control. Thus, this suggests that EVs released after injury are cell type specific and differently react in distinct parts of the body [38]. Different results were found on some cell types, BV2 cells, with the same concentration of LPS in relation to the presence of proinflammatory cytokines in EVs released after LPS stimulation; in fact, according to Yang et al., the EVs concentration of TNF-α and IL-6 after LPS stimulation was increased in comparison to control cells; however, Jones at al. have found that BV2 stimulated with LPS produced EVs packed with a lower concentration of TNF-α, IL-1β and similar concentration of IL-6. The different contrasting results derived from these two studies could be explained by the different stimulation time of the BV2 cells with LPS, which is 12 h in Yang’s [35] study, and 24 h and 48 h in Jones et al. [36] studies. We could postulate that EVs cargo is influenced by the duration of LPS stimulation in BV2 cells. EVs do not only have neuroinflammatory effects; in fact, treatment of neurons with M2 microglia-derived exosomes enhanced miR-124 expression and had a neuroprotective effect by reducing ischemic brain injury and enhancing neuronal survival. The neuroprotective effects of exosomes were reversed when neurons were treated with miR-124 knockdown M2-EXO, demonstrating that the neuroprotective impact of exosomes is related to miR-124 transfer to neurons rather than endogenous expression of neurons [39]. Zhang et al. have also demonstrated that exosomes derived from microglia in M2 phenotype (BV2-Exo) might reduce ischemic brain injury in vitro and in vivo by transferring exosomal miRNA-137 and its targeting gene Notch1 [40]. NF-κB regulates a variety of proinflammatory genes in microglial cells, including iNOS, TNF-α, and IL-6 [41]. Different authors have reported that when activated by stimulants such as phorbol ester and LPS, which are commonly used for activating macrophages and microglia in vitro [42], cultured microglia release various neurotoxic substances such as reactive oxygen intermediates, nitric oxide [43], and cytokines [44], but we could add that EVs released from microglia could also play a role in microglia activation. Our findings showed that EVs-LPS induced morphological changes in microglia that were similar to those induced by LPS, proving that EVs can charge and transport LPS as well as induce morphological changes in BV2 cells. The analysis of the expression of M1-phenotype CD40 and the M2-phenotype CD206 markers is not as simple as it seems and shows that LPS treatment of BV2 cells induces a significant increase of CD40 in comparison to control, and EVs-LPS were able to induce a significant increase of CD40 expression in stimulated cells compared to LPS-stimulated BV2 cells. Regarding CD206, we found that although both LPS and EVs-LPS treatments were able to increase CD206 expression in comparison to controls, no significant differences were detected between LPS and EVs-LPS treatments of BV2 cells. Overall, these findings suggest that EVs-LPS basically tend to drive microglia cells towards a pro-inflammatory activity. Activated microglia approach injured sites in the CNS in many pathological conditions, releasing a variety of inflammatory molecules that exacerbate inflammation and promote neurodegeneration. To mediate the extension of chronic inflammation, activated microglia have a highly motile characteristic. We found that EVs-LPS treatment of BV2 cells increases cell migration in the same way that LPS does, indicating that EVs can activate all of the proteins involved in cell migration. In the present study, MTT assay confirmed that EVs-CTR and EVs-LPS had no effect on cell viability when compared to control cells, indicating that the LPS transported by EVs is not involved in the cell proliferation pathway. TLRs are innate immunity receptors that play an important role in the early stages of immunity. TLR4 is an important pattern recognition receptor that is expressed in microglial cells [45] and is responsible for triggering an inflammatory cascade in microglia when it binds to LPS, which in turn activates several transduction pathways molecules, like JNK and NF-kB [46]. TLR4 expression was higher in BV2 cells treated with EVs-LPS or LPS in comparison to control or EVs-CTR, as evidenced by Western blotting analysis; more probably, EVs-LPS act on BV2 cells trough TLR4 receptor to induce cell activation. The most likely pathway is a binding to the receptor, but future studies will be addressed to describe more in detail this link or activation. Also, the expression of p-AKT and p-IKBα resulted significantly higher after EVs-LPS treatment compared to LPS treatment, thus indicating that EVS-LPS are able to enhance functional pro-inflammatory properties of microglia. Moreover, the release of pro-inflammatory mediators, such as TNF-α, induced by EVs-LPS treatment suggests a role for the NF-κB signaling pathway triggered by EVs-LPS. These results are supported also by reduced release of IL-10, an anti-inflammatory mediator. Our in vitro study demonstrates for the first time that microglial-derived EVs may participate in neuroinflammatory responses by spreading inflammatory messages. The major findings emerging from this study conducted on extracellular vesicles released from LPS-activated cells are: (i) EVs-LPS are able to polarize BV2 microglia cells, (ii) induce migration in BV2 cells, and (iii) induce the release and the expression of pro-inflammatory mediators.

## 5. Conclusions

Altogether, the findings of this study present a novel mechanism underlying neuroinflammation induced by the brain-resident immune cells and their derived EVs, demonstrating that EVs released from LPS-activated and non-activated BV2 microglia possess a different cargo and elicit distinct biological responses in recipient cells. Therefore, EVs play a key role in cell–cell communication, altering the phenotype of recipient cells, and they may be vehicles for agents that propagate neuroinflammation in the CNS. Intriguingly, EVs from microglia may provide new treatment tools for monitoring and diagnosing disease, given their relevance in propagating neuroinflammation. The results of this research were obtained by experiments conducted in vitro, which could represent one of the major limitations; however, they strongly recall further studies on this topic to increase the knowledge of this new cellular communication mechanism in the brain. 

## Figures and Tables

**Figure 1 biology-11-01426-f001:**
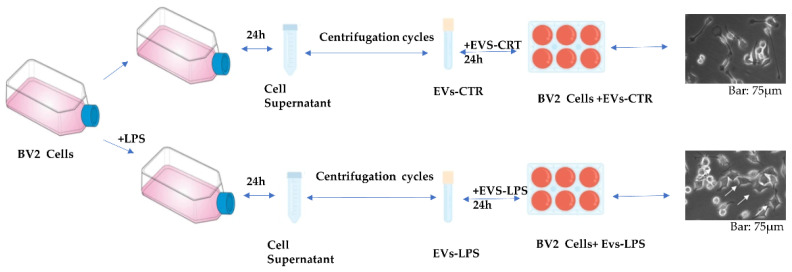
Schematic representation of EVs production without (EVs-CTR) and with LPS (EVs-LPS) stimulation of BV2 cells.

**Figure 2 biology-11-01426-f002:**
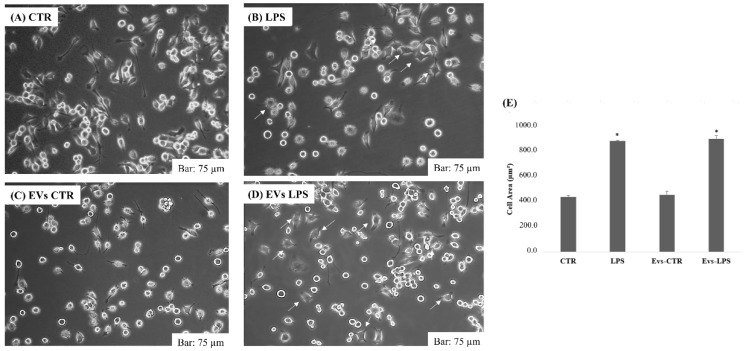
Morphology analysis of BV2 cells cultured in 6-plates in absence of treatment (**A**) or after treatment with LPS (1 ug/mL) (**B**), EVs-CTR (10 μg/mL) (**C**), and EVs-LPS (10 μg/mL) (**D**). Bar 75 µm. (**E**) Cell areas were quantified using Image J software in calibrated unit (µm^2^). * *p* < 0.05 compared with CTR. Data are expressed as means ± SD of three independent experiments.

**Figure 3 biology-11-01426-f003:**
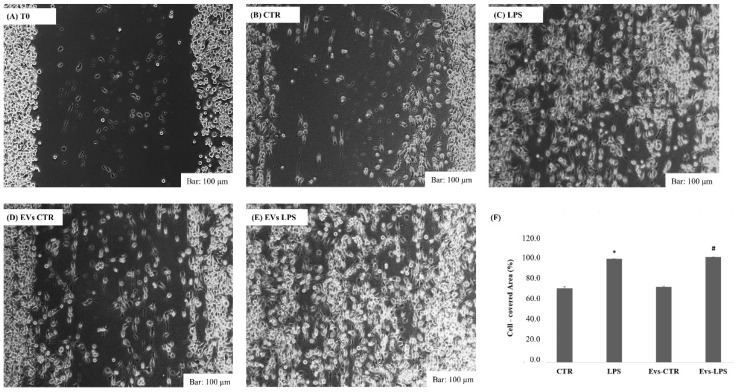
Analysis of cell migration determined by wound healing assay. Representative images of wound healing assay at time 0 (**A**), after 24 h in control cells (**B**), cell stimulated with LPS (1 ug/mL) (**C**), or with EVs-CTR (10 μg/mL) (**D**) or EVs-LPS (10 μg/mL) (**E**). Bar 100 µm. (**F**) Cell areas were quantified using Image J software. Data are presented as the mean ± SD of triplicate experiments. (* *p*-value ˂ 0.05 compared to control cells; # *p*-value ˂ 0.05 compared to LPS).

**Figure 4 biology-11-01426-f004:**
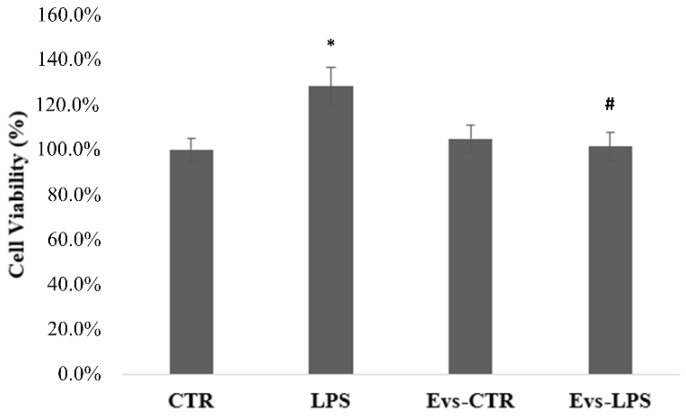
Analysis of BV2 cells proliferation evaluated by MTT assay. Histograms represent the percentage, with respect to control cells (CTR, 100%), of proliferating cells after the exposure to LPS, EVs-CTR, or EVS-LPS. (* *p*-value ˂ 0.05 compared to control cells; # *p*-value ˂ 0.05 compared to LPS). Data are expressed as the mean ± SD of three independent experiments.

**Figure 5 biology-11-01426-f005:**
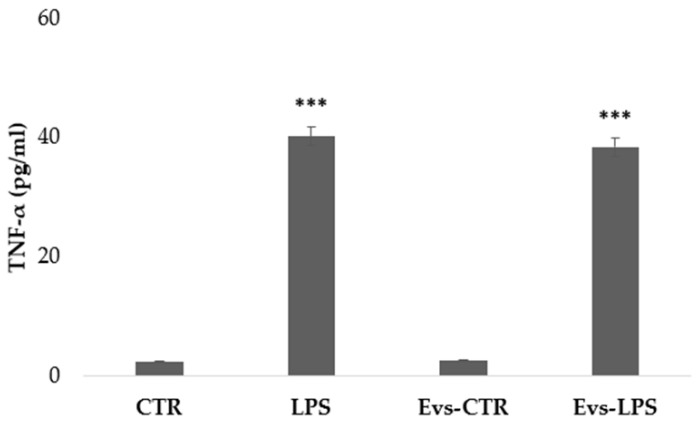
The amount of TNF-α production in supernatant was evaluated by ELISA. TNF-a was increased in BV-2 cells stimulated with LPS and EVs-LPS with respect to control cells (*** *p*-value < 0.001 compared to control cells). Data are expressed as the mean ± SD of three independent experiments.

**Figure 6 biology-11-01426-f006:**
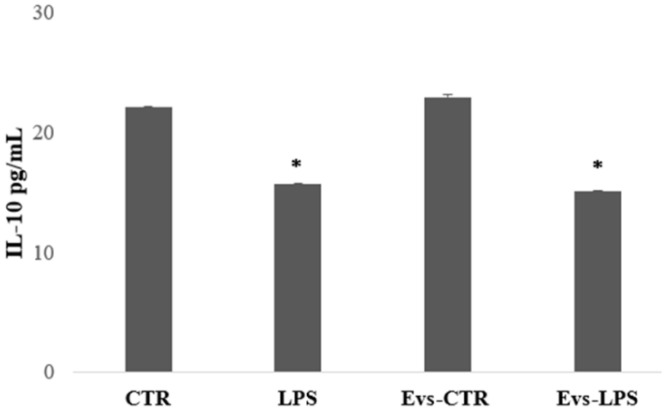
The amount of IL-10 production in supernatant was evaluated by ELISA. IL-10 was increased in BV-2 cells in the control and stimulated with EVs CTR. (* *p*-value < 0.05 compared to control cells). Data are expressed as the mean ± SD of three independent experiments.

**Figure 7 biology-11-01426-f007:**
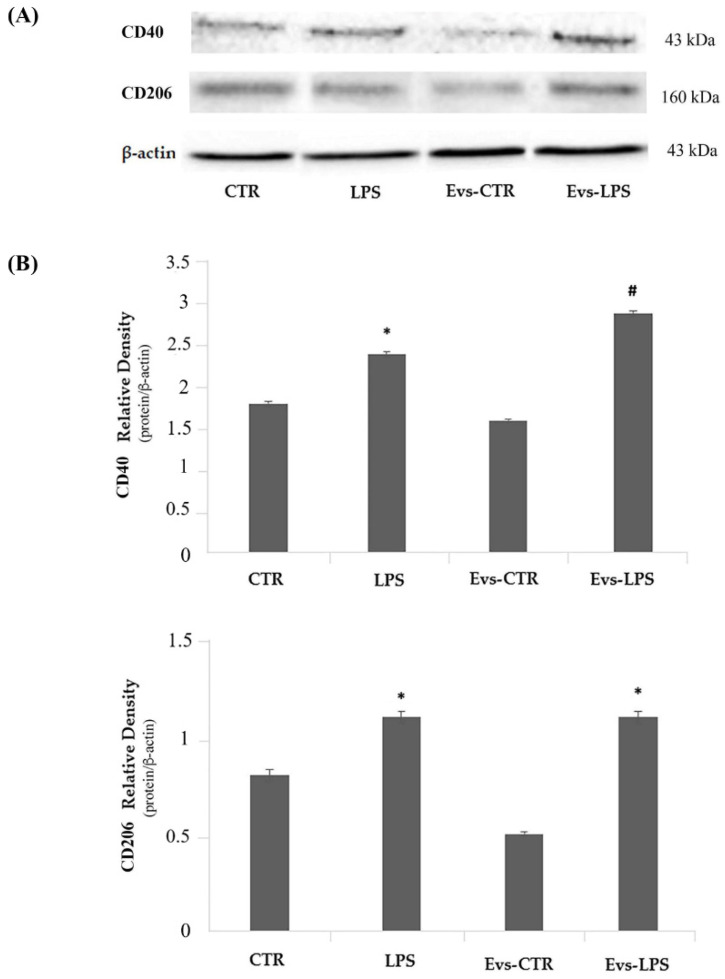
Panel (**A**) Western blotting detection of pro-inflammatory CD40 and anti-inflammatory CD206 in untreated cells (CTR), cells stimulated with LPS, EVs-CTR, and EVs-LPS. Densitomentric analysis of expression levels of the pro-inflammatory CD40 (Panel **B**, upper) and anti-inflammatory CD206 (Panel **B**, lower) * *p*-value < 0.05 compared to the control; # *p*-value < 0.05 in comparison to LPS-treatment. Protein expression levels were normalized to β-actin and results of densitometric analysis are expressed as the mean ± SD of five independent experiments.

**Figure 8 biology-11-01426-f008:**
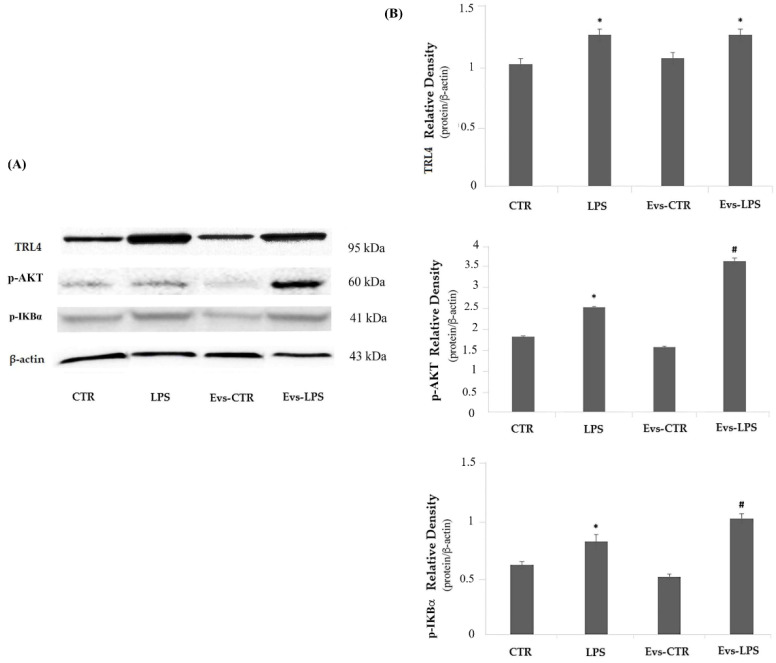
Panel (**A**) Western blotting detection of TLR4, p-AKT and p-IKBα in untreated cells (CTR), cells stimulated with LPS, EVs-CTR, and EVs-LPS. Panel (**B**) Densitomentric analysis of expression levels of TLR4 (**B** upper), p-AKT (**B** middle) and p-IKBα (**B** lower). * *p*-value < 0.05 compared to CTR; # *p*-value < 0.05 compared to LPS; # *p*-value < 0.05 in comparison to LPS. Protein expression levels were normalized to β-actin, and the results of densitometric analysis are expressed as the mean ± SD of five independent experiments.

## Data Availability

Data sharing is not applicable to this article.

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
