# Peer review of "Extracellular Vesicles Cargo in Modulating Microglia Functional Responses"

_biology, 2022, doi:10.3390/biology11101426_

Round 1
Reviewer 1 Report
in the present manuscript, The authors have described Extracellular vesicles (EVs) release due to microglia activation in response to LPS administration in the BV2 cell lines. Also, they showed the naive BV2 activation because of EVs administration, which are collected from previouly LPS-stimulated BV-2 cell lines. The study is novel and gives important insights, but I would like to suggest certain modifications in the revised manuscript.
1- Figure 1, The cells were cropped from the figure 2. It would be good if you can provide corresponding Scale Bars.
2- Please describe the quantitation used in figure 2 for morphology analysis as the presentation of the Cell area is ambiguous to the readers.
3- The significance representation is not appropriate in every figure except figure 5 where ***p-value < 0.001, rest everywhere it is like p-value ˂0,05 and that is not right. Please correct.
4- Please include the value of SD in every figure as it seems very small and I doubt if the authors are calculating Std error and presenting as SD.
5- In figure 7A the quantitation does not match with the representative bands, please provide either good bands or new quantitation.
Author Response
We thank the reviewer, which gave us the opportunity, with its suggestions, to improve the quality of the manuscript and add further useful information on the field. We have carefully taken their comments into consideration in preparing our revision, resulting, as we hope, in a paper clearer, more compelling, and broader. Changes in the manuscript are reported in red.
Below are the answers point by point:
In the present manuscript, The authors have described Extracellular vesicles (EVs) release due to microglia activation in response to LPS administration in the BV2 cell lines. Also, they showed the naive BV2 activation because of EVs administration, which are collected from previouly LPS-stimulated BV-2 cell lines. The study is novel and gives important insights, but I would like to suggest certain modifications in the revised manuscript.
1-Figure 1, The cells were cropped from the figure 2. It would be good if you can provide corresponding Scale Bars.
We thank the reviewer for the suggestion, we have added the scale to the image.
2- Please describe the quantitation used in figure 2 for morphology analysis as the presentation of the Cell area is ambiguous to the readers.
We thank for the suggestion, the ImageJ software calculates the area in calibrated units, such as square micrometers. We have added this information in the text line 254:
"Cell areas were quantified using Image J software in calibrated unit (µm²)."
3- The significance representation is not appropriate in every figure except figure 5 where ***p-value < 0.001, rest everywhere it is like p-value ˂0,05 and that is not right. Please correct.
We thank the reviewer for the suggestion,we have correct the legend of picture 4 and the p-value is <0.001.
4- Please include the value of SD in every figure as it seems very small and I doubt if the authors are calculating Std error and presenting as SD.
The SD has already been added in the graphs, although it looks like a very small number.
5- In figure 7A the quantitation does not match with the representative bands, please provide either good bands or new quantitation.
We thank the reviewer for his suggestion, we have replaced the blot bans in fig 7A with a more representative blot in which the modulation is better highlighted and we checked the densitometric analysis

Reviewer 2 Report
In this manuscript, authors attempted to study the effect of EV's derived from BV2 cells after inducing with LPS and treated microglia with the same to study EV's+LPS effect on microglial structural changes and proinflammatory responses. though the hypothesis seem to be interesting to the field of extracellular vesicles, but importance of this finding to relate with pathology is poorly described or less evident.
Major points:
1. authors used murine micorglia cells for the entire study. they should use human cells like HMC3 to see whether the same phenomena can be observed
2. Typically people use IBA1 as a marker for microglial activation by immunoflourence and western blot. authors didn't try any of those
3. authors should include both pro and anti inflammatory cytokines. just showing TNFalpha isn't sufficient.
4. in Figure 7A, TLR4 blot second lane showing little increase in blot but graph shows increased level. is it the best representative blot? number of biological replicates should be mentioned in the figure legends.
5. Figure 7 A B C, all the actin blots look same. if its the same lysate then it should be made as one figure like common actin and other antibodies. no point in making individual figures.
6. Figure 6 A, actin looks same as Figure 7
Author Response
We thank the reviewer, which gave us the opportunity, with its suggestions, to improve the quality of the manuscript and add further useful information on the field. We have carefully taken their comments into consideration in preparing our revision, resulting, as we hope, in a paper clearer, more compelling, and broader. Changes in the manuscript are reported in red.
Below are the answers point by point:
In this manuscript, authors attempted to study the effect of EV's derived from BV2 cells after inducing with LPS and treated microglia with the same to study EV's+LPS effect on microglial structural changes and proinflammatory responses. though the hypothesis seems to be interesting to the field of extracellular vesicles, but importance of this finding to relate with pathology is poorly described or less evident.
Major points:
- authors used murine microglia cells for the entire study. they should use human cells like HMC3 to see whether the same phenomena can be observed
We thank the reviewer for this suggestion, we will consider HMC3 cells in the next study. In our laboratory at moment we do not possess this cellular line, so it is a little bit difficult for us add these data to this study, but the next step of this research is to verify in in vivo model and in human cell lines, the effects of extracellular vesicles produced after LPS stimulation.
- Typically, people use IBA1 as a marker for microglial activation by immunoflourence and western blot. authors didn't try any of those
We thank the reviewer for the suggestion. However, we point out that, IBA1 protein expression is found in relevant amounts in resting microglia and minor microglia activation might not be detected by cell counting (false negative results are obtained). Moreover, a recent work revealed that Iba-1 protein expression might not be a sensitive marker to evaluate the activation of microglia in an experimental diabetic retinopathy (Shi FJ, et al. Int J Ophthalmol. 2021. PMID: 33614447). We have, therefore, decided to consider the expression of CD 40 as a specific marker of microglial activation because, as reported in literature, the cluster of differentiation (CD) 40 can be assessed as specific markers of microglial activation (Lynch MA, Mol Neurobiol. 2009 PMID: 19629762).
- authors should include both pro and anti inflammatory cytokines. just showing TNFalpha isn't sufficient.
We thank the reviewer for this suggestion. To confirm the result obtained from by TNF-alpha ELISA test, we also evaluated IL-10 ELISA assay. It is known that IL-10 is an anti-inflammatory cytokine and we found a different release of this cytokine compared to TNF-alpha release. We reported it in the text, in material and methods, results and discussion section.
- in Figure 7A, TLR4 blot second lane showing little increase in blot but graph shows increased level. is it the best representative blot? number of biological replicates should be mentioned in the figure legends.
We thank the reviewer for his suggestion, we have replaced with a more representative blot in which the modulation is better highlighted, with the indication in the legend of the number of replicates.
- Figure 7 A B C, all the actin blots look same. if its the same lysate then it should be made as one figure like common actin and other antibodies. no point in making individual figures.
We thank the reviewer for his good suggestion. We proceeded to modify the figure according to the indications received, combining the representations of the blotting using a common b-actin. Moreover, we changed the figure legend according to the new figure.
- Figure 6 A, actin looks same as Figure 7
Many thanks for the right observation. Even in this case we have proceeded to make a single figure using a common b-actin changing, consequently, the legend of the figure.

Round 2
Reviewer 1 Report
The revised manuscript looks improved now. The authors have addressed every question except point 3 in my previous report.
The significance representation is not appropriate in every figure except figure 5 where ***p-value < 0.001, rest everywhere it is like p-value ˂0,05 and that is not right.
Please look carefully for every figure legend and put a dot ( .) instead of a comma (,) while presenting the significant value.
Author Response
We thanks the reviewer, we have revised the statistical analysis and corrected the p-value of the figures.
Reviewer 2 Report
can be accepted in present form
Author Response
We thank the reviewer